



# PEATGRIDS: Mapping thickness and carbon stock of global peatlands via digital soil mapping

Marliana T. Widyastuti[1], Budiman Minasny[1], José Padarian[1], Federico Maggi[2], Matt Aitkenhead[3], Amélie Beucher[4], John Connolly[5], Dian Fiantis[6], Darren Kidd[7], Yuxin Ma[8], Fraser Macfarlane[3], Ciaran Robb[3], Rudiyanto[9], Budi I. Setiawan[10], Muh Taufik[11]

[1]School of Life and Environmental Sciences, The University of Sydney, Eveleigh NSW 2015, Australia
[2]School of Civil Engineering, The University of Sydney, Darlington NSW 2008, Australia
[3]The James Hutton Institute, Craigiebuckler, Aberdeen, AB15 8QH Scotland, UK
[4]Department of Agroecology, Aarhus University, Blichers Allé 20, 8830 Tjele, Denmark
[5]School of Natural Sciences, Trinity College Dublin, Dublin 2, Ireland
[6]Department of Soil Science, Faculty of Agriculture, Universitas Andalas, Padang, Indonesia
[7]Department of Natural Resources and Environment Tasmania, Prospect, Tasmania 7250, Australia
[8]Landcare Research, Palmerston North 4442, New Zealand
[9]Program of Crop Science, Faculty of Fisheries and Food Science, Universiti Malaysia Terengganu, Kuala Nerus 21030, Malaysia
[10]Department of Civil and Environmental Engineering, IPB University, Bogor 16680, Indonesia
[11]Department of Geophysics and Meteorology, IPB University, Bogor 16680, Indonesia

*Correspondence to*: Marliana T. Widyastuti (marlianatri.widyastuti@sydney.edu.au)

**Abstract.** Peatlands, which only cover 3 to 5 percent of the global land area, can store up to twice the amount of carbon as the world's forests. Although recognised for their significant role in the global carbon cycle, discovering the global extent of peatlands and their carbon stock remains challenging. Referring to the UNEP's global peatland map, here we present PEATGRIDS, a data product containing global maps of peat thickness and carbon stock created created using the digital soil mapping approach. We compiled over 25,000 observations of peatland thickness, bulk density (BD) and carbon content (CC), globally. Using the Random Forest (RF) algorithm, we estimated peat thickness and peat BD and CC at ~1 km resolution at multiple depths (0 – 2 m) globally. The estimates were generated using 19 land surface covariates from digital maps and remote sensing images of land use, soil characteristics, topographical features, and climate parameters. The RF models for peat thickness were trained on 25,200 points grouped into six geographic regions. Validation of the peat thickness estimates showed a good performance, with the coefficient of determination ($R^2$) ranging from 0.15 to 0.72. The prediction for peat BD and CC followed the same model architecture and were trained on 17,000 and 7,000 points, respectively. Overall, BD and CC models performed well and consistently across soil layers with average $R^2$ values of 0.61 for BD and 0.48 for CC. Based on the estimated peat thickness, BD and CC, the carbon stock of global peatland was estimated to be 1,029 Pg C for peat dominated area of 6.57 million km$^2$. PEATGRIDS is made available at https://doi.org/10.5281/zenodo.12559239 (Widyastuti et al., 2024) to support further analyses and modelling of peatlands across the globe.





## 1 Introduction

Peatlands are critical ecosystems because they store vast amounts of carbon (UNEP, 2022; Leifeld and Menichetti, 2018). Despite covering only 3 to 5% of the Earth's land area, peatlands can store twice as much carbon as all the global above-ground biomass combined (Beaulne et al., 2021; Temmink et al., 2022). Carbon accumulation in peatlands has occurred for thousands of years as dead plant materials are preserved in these water-logged ecosystems. However, given their critical role in the global carbon cycle, peatlands face detrimental degradation due to natural disturbances (Page and Hooijer, 2016) and

anthropogenic activities (Taufik et al., 2020; Joosten, 2009; Fluet-Chouinard et al., 2023). Actions on protecting and restoring peatlands are essential for ensuring the continued provision of its ecosystem services and achieving the Sustainable Development Goals (United Nation, 2015).

Monitoring the extent and condition of peatlands has been challenging due to their complex formation, which involves diverse vegetation, hydrology, climate and geomorphology (Minasny et al., 2023; Holden and Connolly, 2011). Peatlands exist in cool

or wet climates, ranging from the high latitudes in the Northern and Southern Hemisphere to the Tropics. The recent global peatlands assessment (GPA) by United Nations Environment Program (UNEP) reported an updated global peat coverage, reaching up to 500 million ha by defining peatlands as areas with more than 30 cm of peat layer (UNEP, 2022). The GPA also included a Global Peatland Map (GPM) created based on a combination of national and regional data through a plausibility check and incorporated a mapping process based on satellite images. According to the GPM, the global peatland area reached

4.9 million km$^2$, surpassing previous peat extent estimates from geospatial information collection (Immirzi et al., 1992; Page et al., 2011; Xu et al., 2018) and machine learning estimates (Melton et al., 2022).

Modelling the status of peatlands is crucial for understanding their role in the global carbon cycle. They can either be carbon sources, through net peat degradation, or carbon sinks, through net peat preservation and accumulation (Joosten et al., 2016). Mapping and modelling peatland dynamics can assist scientists predict how human pressures and climate change could affect

these ecosystems, influencing global carbon dynamics and climate.

Machine learning approaches have increasingly been used to estimate peat extent and status, mostly on a regional to local scale (Minasny et al., 2019). Several works have been done utilising machine learning algorithms to create detailed maps of peatland status, such as across Ireland (Habib and Connolly, 2023; Habib et al., 2024a; Habib et al., 2024b). Random forest models were used to estimate the carbon stock of northern peatlands. This region has the most comprehensive peat coverage, estimated

to hold 415 Pg C across 370 million ha, with nearly half being permafrost affected (Hugelius et al., 2020). In the Congo Basin, peatlands were mapped using the random forest model, revealing 29 Pg C stored in 16.7 million ha, with only 8% of peatlands lying within a nationally protected area (Crezee et al., 2022). Furthermore, a neural network model was used to map peat coverage over Scotland based on remote sensing data combined with other spatial covariates (Aitkenhead and Coull, 2019). Several machine learning methods were also tested to estimate carbon loss due to oil palm plantations in a peatland of 54,000

ha in West Sumatra, Indonesia (Fiantis et al., 2023). In addition, machine learning approaches offer rapid and lower-cost map production with relatively good accuracy compared to traditional mapping techniques (Rudiyanto et al., 2018).

While there have been several efforts to create and update global peat extent (Xu et al., 2018), information about peatland status, particularly the carbon stock, is still lacking. Global soil mapping efforts have not accounted for the differentiation of peatlands and their thicknesses (Poggio et al., 2021). Thus, this paper aims to introduces PEATGRIDS, a data product

providing maps of global peat thickness and carbon stock derived from the digital soil mapping approach. The specific objectives are: (1) to evaluate random forest models in predicting thickness, bulk density, and carbon content for peatlands with the help of spatial covariates, and (2) to generate 1 km raster prediction maps for peat thickness and carbon stock across the globe. We limited our mapping effort to areas considered as peat according to the GPM (UNEP, 2021). This work provides a comprehensive global maps of peatlands and analyses aiding the global modelling of peatland status.

**2 Methods**

**2.1 Global peat extent**

The geographical area of interest in PEATGRIDS product is the one considered potentially as peatlands according to the Global Peat Map (GPM) version 2.0 (UNEP, 2021), obtained from the Global Peatland Database (https://greifswaldmoor.de/global-peatland-database-en.html, accessed in April 2024). The GPM data was compiled by the

Greifswald Mire Centre following a 'bottom-up' approach, which combined more than 200 geo-datasets including fill data gaps. The GPM, available at 1-km resolution, reports up to 8.7 million $km^2$, double the peat area of a previous assessments (Xu et al., 2018). We limited our prediction to 'peat dominated' lands, covering 6.7 million km² or 5% of the earth's land area. We assume that the peat-dominated areas are peatlands due to the fact that no information about peat thickness was found in the GPM. Since it is notable that the GPM overestimated Indonesian peatlands, we specified the extent of Indonesian peat

using the national map available from the Indonesian Ministry of Agriculture (Haryono et al., 2011). This reduces the overall global peat coverage to 6.57 million km².

To expedite global modelling and consider different climatic zones, we divided the global peat map into six regions based on continents and climate zones (Fig. 1).

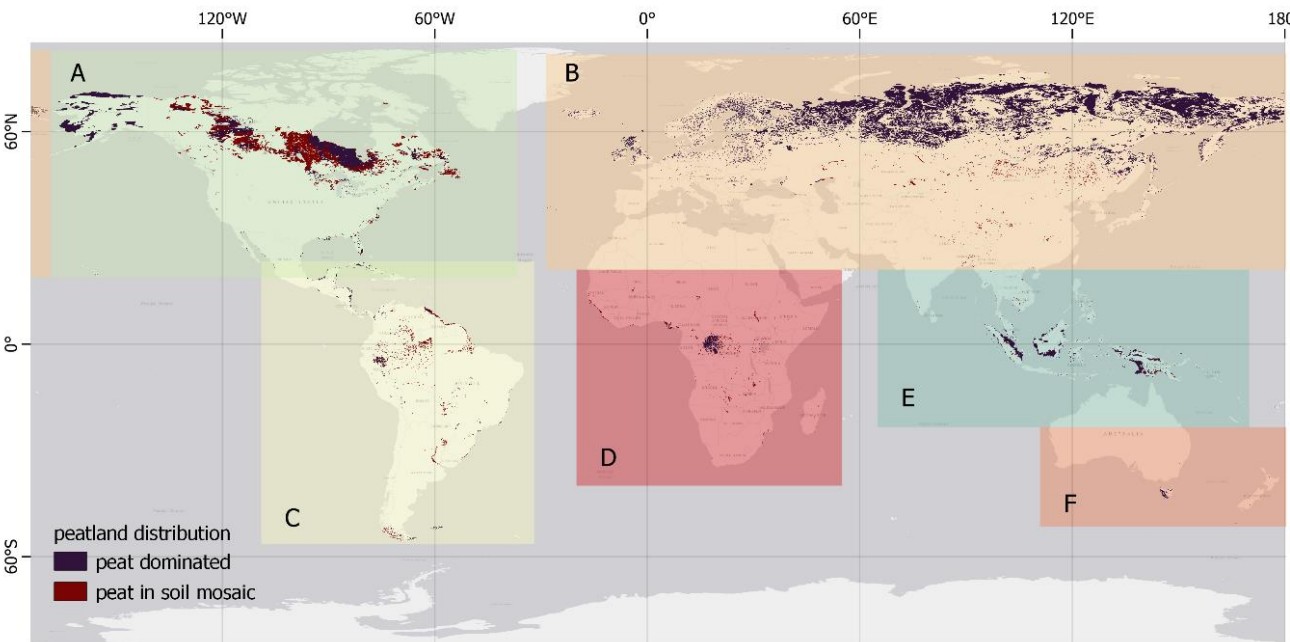

**Figure 1: Distribution of global peatlands according to the Global Peatland Map version 2.0. The global map was divided into six regions for specific analysis in this work (Region A—North America, B—Europe and Russia, C—Latin America, D—Africa, E—Southeast Asia, F—Australia and New Zealand).**

## 2.2 Data collection

We gathered peat observation data from multiple resources, including published databases and research papers, listed in Table S1 (Supp. Material). For peat thickness data, we used available peat thickness maps to complement existing point measurements, especially for regions lacking field data. We sampled points on the maps randomly using *sampleRandom* function in R and treated them as observations. In Indonesia, we extracted 300 points in total based on the national map of peat depth classes, which grouped peat thickness into six categories (Anda et al., 2021). We took the mid value from each class and set 9 m as the maximum peat thickness. We followed the same procedure for some European nations where data points are not publicly available, i.e. the Netherlands, Sweden, and Denmark. The peat map of Sweden provides information on peat coverage across the country according to certain peat depth values as a threshold, thus we assumed the threshold value as the peat thickness. While the sampled data may have errors/uncertainties, this approach will help reduce the final model bias.

For peat BD and CC, we considered the range of soil depth intervals at which the measurements were taken. If the data were given at specific depths, we assumed that these measurements represent the interval 5 cm above and below the measurement level. If explicit measurement depths were not provided, but peat thickness at the same location was available, we assumed that the peat BD and CC represented the average value for that peat thickness.

To enrich the spatial representation of the data, we estimated BD and CC values based on loss-on-ignition (LOI) data, which represents the amount of organic matter contained within the soil, using pedotransfer functions. We multiplied the LOI value



by 0.58 to calculate carbon content based on the van Bemmelen factor (Van Bemmelen, 1890; Minasny et al., 2020). For bulk

110    density, we used the logarithmic functions of CC for surface and subsurface soil layers developed by Harrison and Bocock (1981). We calculated carbon density (in Mg m⁻³) by multiplying the BD and CC when both measurements were available at a single point. Additionally, some databases provided geo-referenced carbon density data directly.

Data on peat thickness, BD, or CC, along with their geographical coordinates, were compiled into one database. To harmonise the dataset, we standardised the measurement units for peat thickness in meter (m), BD in Mg m⁻³, and CC in mass fraction (g

115    g⁻¹). Prior to modelling, the data was carefully checked and filtered to ensure data reliability. Specifically, we excluded BD values more than 1 Mg m⁻³, and only included CC within threshold 0.1 – 0.58 g g⁻¹ for further analysis. Regarding the various measurements depth intervals, we interpolated the values throughout peat layers into five layers (0-15 cm, 15-30 cm, 30-60 cm, 60-100 cm and 100-200 cm depth) using the mass-preserving spline function (*mpsline2*) in R language (Bishop et al., 1999). In cases where multiple data points located within the same pixel of our target resolution (1 km x 1 km), an average

120    value was considered. This aggregation was performed using *rasterize* function in R based on a 1 km global raster data. The spatial distribution of peat data points for each property is shown in Fig.2.

**Figure 2: Location of peat data points for each property. Colour map shows the number of points on a natural logarithmic scale existing within the area of ~100 km x ~100 km (1º x 1º resolution).**



### 2.3 Modelling scheme

The random forest (RF) algorithm was employed to build regression models for estimating peat thickness, BD and CC, with the flowchart is shown in Fig.3. The model represents the average of multiple decision trees constructed using a subset of covariates and data points. We used 19 covariates (Table 1) representing peat formation factors to predict peat thickness, BD, and CC separately. We selected remotely sensed imagery products that are relevant to the presence of peatlands. All covariates obtained from the literature and the Google Earth Engine (GEE) database were harmonised to a 1 km spatial resolution using nearest neighbor resampling approach. The rasters were in the WGS84 coordinate reference system (EPSG=4326) covering spatial extent of 180°E-180°W 90°N-90°S. Field observations were paired with corresponding covariates. We then exported these data and raster of covariates from GEE and performed RF modelling and mapping in Python.

The RF models were trained following a five-repeated calibration scheme against the observations data. The data were randomly split into a training set (70%) and a testing set (30%). During the training process, we applied fivefold cross-validation (CV) to fine tune the RF hyperparameters, including the number of trees, number of features, number of samples, and minimum number of leaves (See Table S2 in Supp. Material). The tuning process used the 'bagging' method to generate trees and measured the out-of-bag data samples to calculate the accuracy scores for each trial. The process was executed using the *RandomizedSearchCV* function with a total of 800 fitting models covering 160 combinations of hyperparameter values. We selected the hyperparameter values with the highest cross-validation score as the final model. For each repetition, the final model was evaluated on the testing dataset to gauge the model's performance.

Model performances were quantified based on root mean square error (RMSE) and the coefficient of determination ($R^2$). The RMSE measures the standard deviation of the prediction errors relative to observations, while the $R^2$ represents how well the models can capture the variation of the dependent variable. The mathematical equations to calculate $R^2$ and RMSE are given in Eq. (1) and Eq. (2):

$$R^2 = 1 - \frac{RSS}{TSS} \tag{1}$$

$$RMSE = \sqrt{\frac{\sum_{i=1}^{N}(y_i - x_i)^2}{N}} \tag{2}$$

where RSS is the sum squares of the bias value of the prediction to its linear regression, while TSS is the total sum squares of the mean error of prediction. $N$ is the number of data points, $y_i$ and $x_i$ are our model prediction and observation values, respectively.



**Table 1: List of environmental covariates to develop model predictions of peat thickness, bulk density, and carbon content available globally derived from Google Earth Engine Database and literatures.**

| Dataset | Bands | Unit | Spatial resolution | Description | References |
|---|---|---|---|---|---|
| Geomorpho90m | slope | - | ~90m | The rate of change of elevation in the direction of the water flow line. | Amatulli et al. (2020) |
| | cti | - | ~90m | Compound topographic index, or topographic wetness index, is computed as the logarithm of the cumulative upstream catchment area divided by the tangent of the local slope angle. | Beven and Kirkby (1979) |
| | geom | - | ~90m | Geomorphological forms: flat, peak or summit, ridge, shoulder, spur, slope, hollow, foot slope, valley, and pit or depression. | Jasiewicz and Stepinski (2013) |
| | roughness | - | ~90m | The largest inter-cell absolute difference of a focal cell and its 8 surrounding cells. | Beasom et al. (1983) |
| | spi | - | ~90m | Stream power index. the erosive power associated with flow and the tendency of gravitational forces to move water downstream. | Moore et al. (1991) |
| Copernicus Digital Elevation Model (GLO-30 DEM) | dem | m | ~30m | The surface of the Earth including buildings, infrastructure, and vegetation. | European Space Agency (2021) |
| Sentinel-1 SAR GRD | VV | dB | ~10m | Single co-polarization, vertical transmit/vertical receive. | |
| | VH | dB | ~10m | Dual-band cross-polarization, vertical transmit/horizontal receive. | |
| WorldClim | tmin | °C | ~1km | Average monthly minimum temperature the year of 1960-1991. | Hijmans et al. (2005) |
| | tmax | °C | ~1km | Average monthly maximum temperature the year of 1960-1991. | |
| | prec | mm | ~1km | Average monthly precipitation for the year 1960-1991. | |
| MCD12Q1.061 MODIS Land Cover Type | LC_Type1 | - | ~500m | Global land cover types at yearly intervals. | Friedl and Sulla-Menashe (2022) |
| Global PALSAR-2/PALSAR Yearly Mosaic, version 2 | HH | - | ~25m | HH polarization backscattering coefficient, 16-bit DN. | Shimada et al. (2014) |
| | HV | - | ~25m | HV polarization backscattering coefficient, 16-bit DN. | |
| VNP13A1: VIIRS Vegetation Indices 16-Day 500m | EVI2 | - | ~500m | second band Enhanced Vegetation Index. | Didan and Barreto (2018) |
| | NIR_reflectance | - | ~500m | Near-infrared Radiation reflectance. | |
| Soil Grids v2.0 | soc_mean | g kg$^{-1}$ | ~250m | Soil organic carbon content in the fine earth fraction. | (Poggio et al., 2021) |
| Global Pattern of Groundwater Table Depth | | m | ~1km | A groundwater model-based data based on global archives and literatures. | (Fan et al., 2013) |
| Thickness of Soil | | m | ~1km | A modelled-based data based on available topography, climate, and geology map as input. | (Pelletier et al., 2016) |





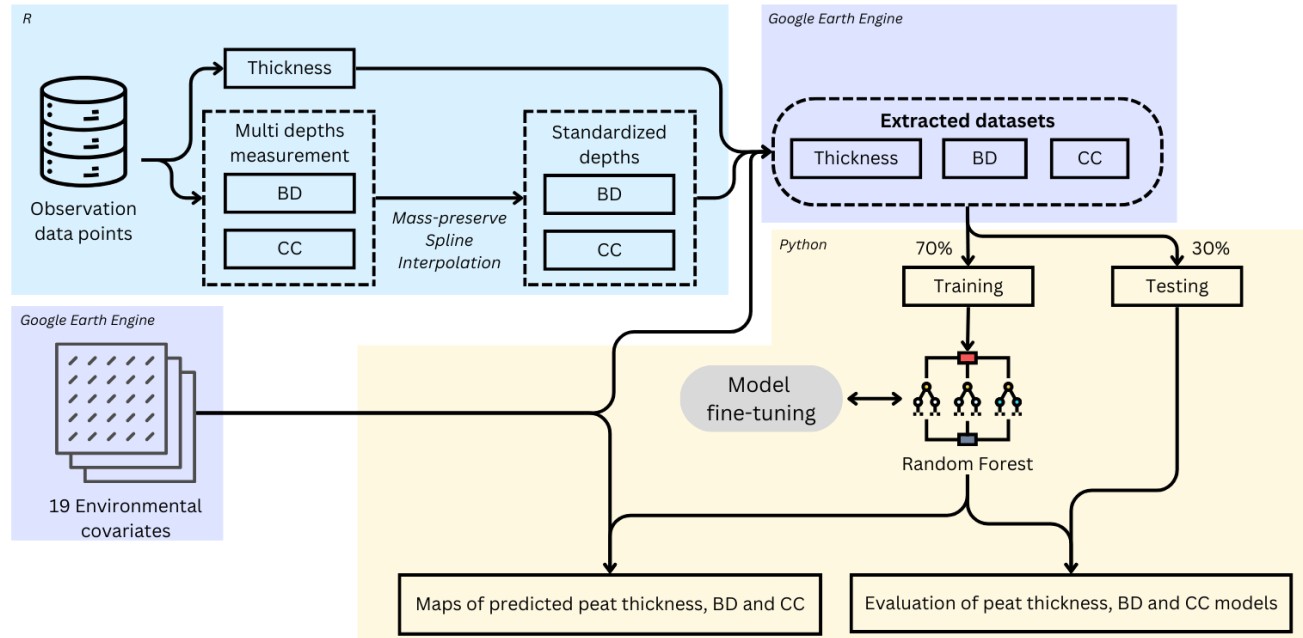

**Figure 3: Modelling schema to generate predictions of peat carbon stock over the global peatlands area. BD=bulk density, CC= carbon content.**

In addition, we also investigated the map accuracy by comparing the predicted value of carbon density (Cd) (see Section 2.4) to observation data. The observed data is the averaged Cd across any measurement depth within a 1 km pixel. The predicted Cd is the averaged Cd value across 2 m peat layers extracted from our BD and CC prediction maps at the associated point.

**2.4 Development of global peat thickness and carbon stock map**

The calibrated RF models were extrapolated to the global peatland area using the GPM. Maps of estimates were generated by employing the five-fold trained models on stacked raster data of the covariates. The distributed maps were the mean of five map predictions. Models to predict peat thickness, BD and CC followed the Fig.3 schema.

*Peat thickness map*

Models to estimate global peat thickness were individually trained in six different regions (Fig. 1). Regional maps were then integrated to create the final global peat thickness map. For estimates in the intersection area (North America (A) and Latin America (C) regions as shown in Fig. 1), we selected the prediction from the Latin America models, considered to be more representative of peat thickness for that region.

*Spatial prediction of peat bulk density and carbon content*

Multi-depths of bulk density and carbon content maps were derived from individual RF models for each of five soil layers. Considering the non uniform distribution of data points and insufficient data for modelling in some regions (e.g. Africa, Australia, and New Zealand, see Fig. 2), we trained the RF model based on the global dataset and then extrapolated them to



each region's raster data. We generated maps for each property at each soil depth interval covered by the GPM. These maps
were then used to calculate carbon stock on the grid cell basis.

*Peat carbon stock map*

Carbon stock calculation was based on predicted maps of peat thickness, and the multi-depths of peat BD and CC. We uploaded
all the maps into the GEE environment to perform calculation on 1-km raster data at the global scale. We first calculated the
total carbon stock ($C_{stock}$ in Mg C) for each soil layer as:

$$C_d = CC \times BD \tag{3}$$

$$C_{stock} = C_d \times d_{peat} \times A_{pixel} \tag{4}$$

where $CC$ is carbon content in dry soil (g C g$^{-1}$), $BD$ is bulk density (Mg m$^{-3}$), $C_d$ is carbon density (Mg C m$^{-3}$), $d_{peat}$ is peat
layer thickness (m), and $A_{pixel}$ is area of the raster grid cell (m$^2$). We derived grid cell area information from
*ee.Image.pixelArea* function in the GEE programming language.

Carbon stock for all depths was calculated on a per-grid cell basis, involving the sum of carbon stock for each peat layer up to
the maximum depth present at that grid cell. If the maximum depth exceeds 2 m, we assumed that the deeper peat layer had
the same BD and CC values as predicted at a depth of 100-200 cm.

## 3 Results

### 3.1 Statistics of the global peat dataset

The aggregated 1 km global peat thickness dataset was split by the regions shown in Fig. 1, and the BD and CC datasets by
their standardised layers (Fig. 3).

For peat thickness, the largest number of data was collected in Europe and Russia, comprising more than 70% of the total
sample (n = 25,874). The datasets of six regions were, overall, skewed to the right, indicating that more observations were
lower than the mean value (Fig. 4). Latin America and Australia-NZ were dominated by peat thickness of less than 0.5 m,
resulting in a mean value of less than 1 m (0.42 and 0.83 m, respectively). With the least observed data, Africa had a distribution
of peat thickness ranging between 0.02-5.86 m. Other regions exhibited similar data distribution patterns, with a mean value
of around 2 m, with Southeast Asia having the deepest average peat thickness of 2.97 m.

The bulk density dataset had a larger sample size than the CC dataset across all soil layers. However, the data points were
predominantly located within the Western Siberia region (Fig. 2), accounting for more than 90% of the total data. Overall, the
dataset exhibited a similar distribution across the soil layers, which was skewed to the right, with the highest frequency
observed in the bulk density range of 0.1-0.15 Mg m$^{-3}$. For the 15-30 cm dataset, despite having the largest number of samples
(more than 17,000 points), it had the largest mean bulk density value (0.3 Mg m$^{-3}$), with more than 6000 observations exceeding
0.6 Mg m$^{-3}$.

Despite having the same range of value across the layers, the CC dataset had a distinct distribution (Fig. 4). In addition, the

quantity of data points decreased with depth. The dataset for the 0-30 cm depth, recorded by more than 7,000 data points, was

dominated by low carbon content (< 0.15 g g$^{-1}$), followed by the highest carbon content (> 0.55 g g$^{-1}$) with the remaining data

distributed across carbon content between these values. This led to a lower mean value of carbon content for this layer (0.31 g

g$^{-1}$ and 0.35 g g$^{-1}$ for peat layers 0-15 cm and 15-30 cm, respectively). The dataset was skewed to the left for deeper layers,

increasing the mean value to more than 0.42 g g$^{-1}$.

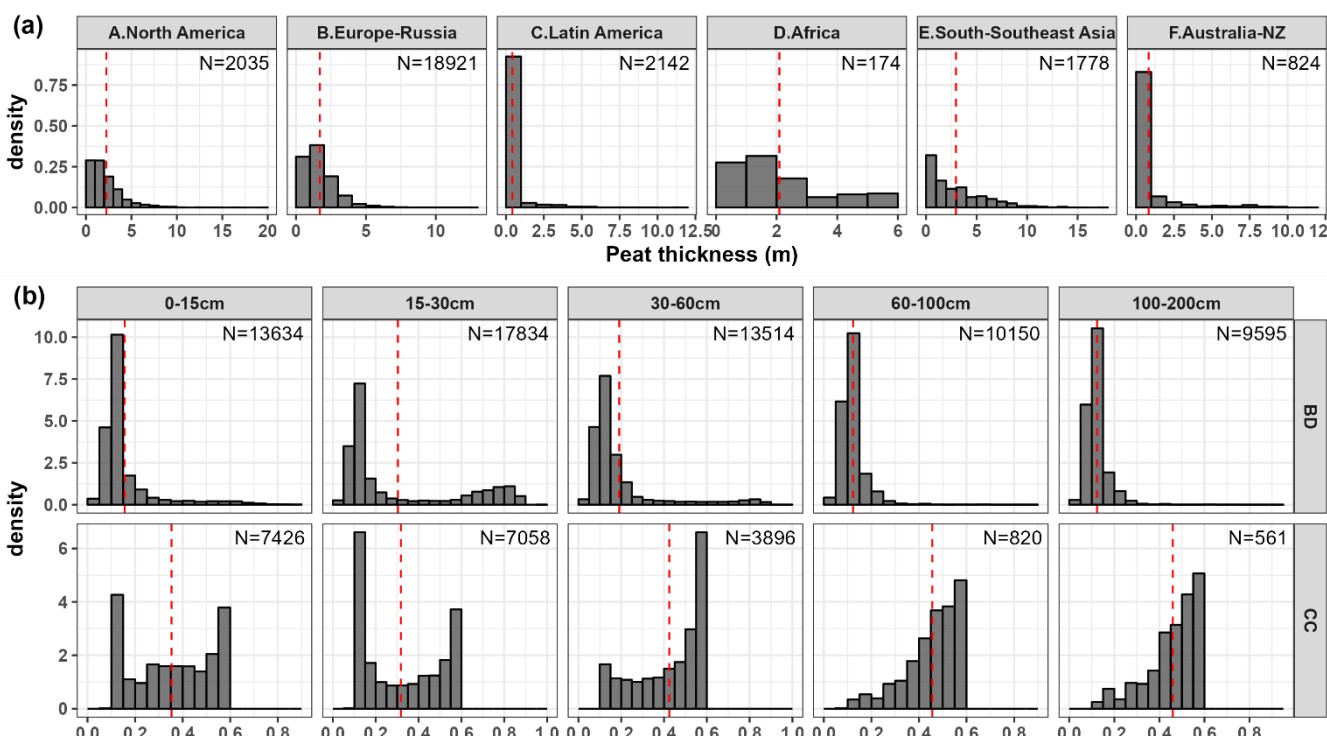

**Figure 4: Histogram of peat data points across the globe: (a) peat thickness data divided into six regions, (b) peat bulk density (BD, in Mg m$^{-3}$) and carbon content (CC, in g g$^{-1}$) at multiple depths. The width of each bar in the histogram for peat thickness represents data of 1 m, yet for peat BD and CC are groups of 0.05 value. The red dashed line shows the mean value.**

**3.2 Performance of the models**

We evaluated the performance of our models using the testing dataset. Table 2 shows the accuracy results based on the averaged

root mean square error (RMSE) and coefficient of determination (R$^2$). Our models performed variably well, with R$^2$ ranging

between 0.15 – 0.80.

The thickness models for Australia-NZ had the highest coefficient of determination (R$^2$ = 0.72) and a RMSE value of less than

1 m. The North America models had the lowest R$^2$ of 0.15, despite having good performance during training (R$^2$ = 0.65, see

Table S3 in Supp. Material). Peat thickness models for the European and Russian regions, which were trained on the largest



dataset, performed better than the North American models, resulting in a lower error (RMSE = 0.96 m) and an $R^2$ of 0.37. Models for Africa and South America resulted in $R^2$ values around 0.5 with RMSE between 0.6 and 1.9 m.

The BD models consistently performed well across all depths, with $R^2$ values above 0.5 and RMSE ranging from 0.04-0.12
Mg m$^{-3}$. The BD models for the 15-30 cm depth layer, which had the largest number of observations among the layers, resulted in the highest averaged $R^2$ value ($R^2 = 0.80$) but also had the largest RMSE of 0.12 Mg m$^{-3}$.

More varied results were observed in the performance of CC models across all layers. Models for the top layer (0-15 cm) had the highest accuracy, with $R^2 = 0.71$ and RMSE = 0.09 g g$^{-1}$. The performance of CC models varied for the peat layer deeper than 15 cm. For the two deepest layers, the models performed better than those for the two layers above. The 30-60 cm depth
models had the lowest accuracy with $R^2 = 0.30$ and RMSE = 0.13 g g$^{-1}$.

In addition, we observed that the importance level of each input feature in our RF models being varied across the regions and the depths (Figure S1 in Supp. Materials). Elevation and climate showed a high importance on the thickness models, yet only for Latin America and Southeast Asia. In BD models, all climate variables (air temperature and annual rainfall) consistently had higher importance than other features; additionally, vegetation indices also had a relatively high impact on the models for
0-15 cm and 15-30 cm.

**Table 2: Prediction accuracy of random forest models for each parameter on a testing dataset. The value for each goodness metrics reflects the average accuracy of multiple models from the modeling scheme. *n* is the amount of data for testing, RMSE = root mean square error (in parameter's unit), $R^2$ = coefficient of determination (unitless).**

| Parameters | Unit | Peatland region | n | RMSE | $R^2$ |
|---|---|---|---|---|---|
| **Thickness** | **m** | A—North America | 611 | 1.68 | 0.15 |
| | | B—Europe and Russia | 5,676 | 0.96 | 0.37 |
| | | C—Latin America | 643 | 0.58 | 0.51 |
| | | D—Africa | 52 | 1.03 | 0.55 |
| | | E—South and Southeast Asia | 533 | 1.91 | 0.51 |
| | | F—Australia and New Zealand | 247 | 0.85 | 0.72 |
| | | **peat layers (cm depth)** | | | |
| **Bulk density** | **Mg m$^{-3}$** | 0 - 15 | 4,090 | 0.08 | 0.54 |
| | | 15 - 30 | 5,350 | 0.12 | 0.80 |
| | | 30 - 60 | 4,054 | 0.11 | 0.63 |
| | | 60 - 100 | 3,045 | 0.04 | 0.55 |
| | | 100 - 200 | 2,871 | 0.04 | 0.56 |
| **Carbon content** | **g g$^{-1}$** | 0 - 15 | 2,228 | 0.09 | 0.71 |
| | | 15 - 30 | 2,117 | 0.11 | 0.62 |
| | | 30 - 60 | 1,169 | 0.13 | 0.30 |
| | | 60 - 100 | 246 | 0.09 | 0.35 |
| | | 100 - 200 | 168 | 0.09 | 0.44 |

### 3.3 Evaluation of carbon density prediction

We further tested our overall prediction accuracy by comparing our estimated carbon density (Cd, calculated from predicted BD multiplied by predicted CC) with the observed Cd values (5,091 data points). Figure 5 shows that our prediction can only capture 18% of the variation in the observed carbon density ($R^2 = 0.18$). Despite the positive linear relationship between the predicted and observed data as shown by the regression line, our predictions mostly overestimated carbon density in the range of 0.05-0.10 Mg C m$^{-3}$, lowering the fitness between the regression line and the 45-degree line.

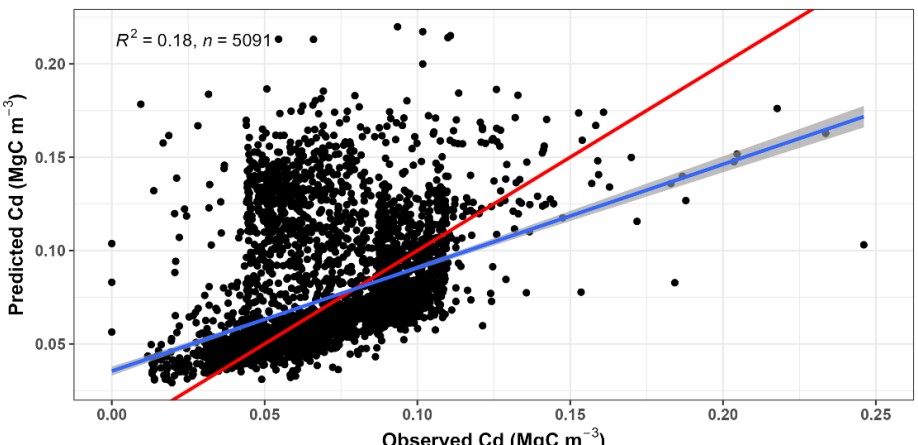

**Figure 5: The agreement between predicted and observed carbon density across global data points. The red line represents 1:1 plot, and the blue line is the regression line.**

### 3.4 Predicted peat thickness and carbon stock

We employed the trained models on 19 covariates to generate maps of peat properties, which were later used to calculate carbon stock by grid cell. We divided the global peat maps into Northern, Tropical, and Southern peatlands based on the 23.5° North and South latitude lines, which is the maximum tilt of the Earth that distinguishes tropical zone, for further analysis.

#### 3.4.1 Northern peatlands

Peat thickness in the Northern region was predicted to range from 0.13 to 5.78 m, with a mean value of 2.14 m. Shallow peats (< 1 m) were found in northern Alaska and European countries, including the UK and Ireland (Fig. 6). Canadian peatlands were mainly predicted to be more than 2 m thick, with shallower peats in the Hudson Bay Lowlands (HBL). In Russia, deeper peats (> 3 m) were predicted in West Siberian Lowlands and some patches in the European region.

Carbon stocks in Northern peatlands varied from 170-7,300 Mg C ha$^{-1}$, with no clear pattern from north to south given the difference climates. Russian peatlands, having the largest peat coverage in the region, were dominated by high carbon stock with more than 1,000 Mg C ha$^{-1}$. Low carbon stocks, less than 500 Mg C ha$^{-1}$, were predicted in northern Alaska and some parts of the Siberian peatlands (Fig. 6). Canadian peatlands were predicted to have carbon stock values of more than 1,000 Mg



C ha$^{-1}$ across the regions. Only a small part of the HBL area had carbon around 800 Mg C ha$^{-1}$. Despite having shallower peats, peatlands in Ireland, the UK and Europe had between 1,000-2,000 Mg C ha$^{-1}$.

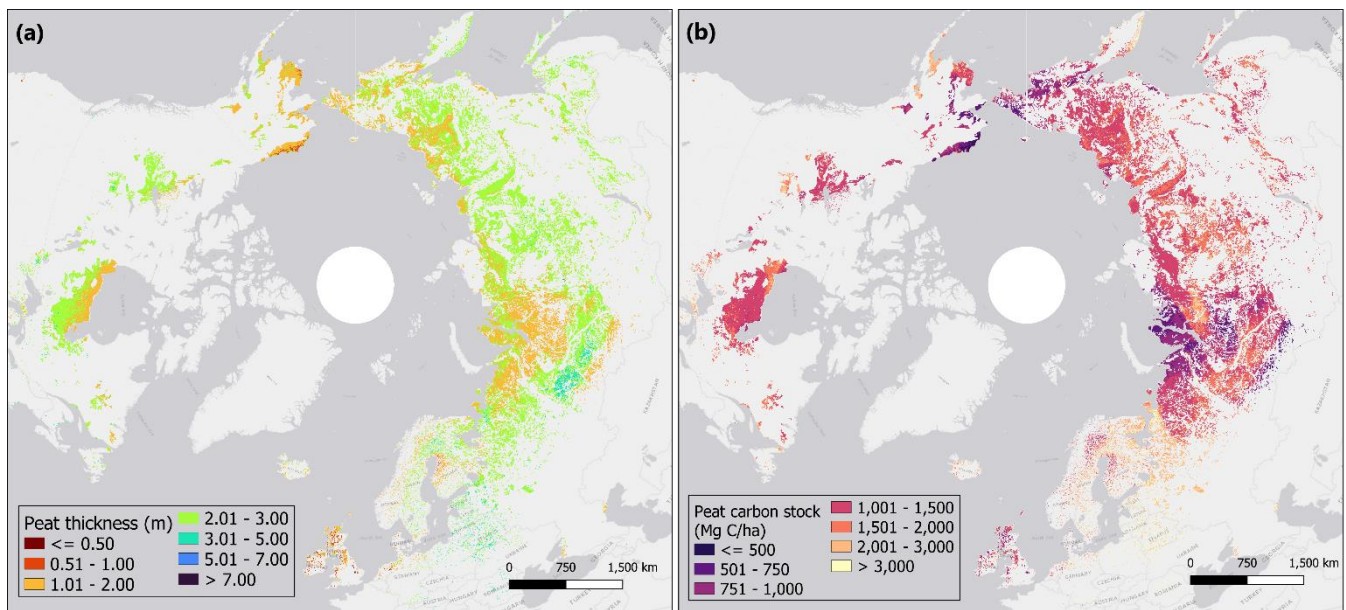


**Figure 6: Maps of (a) peat thickness and (b) carbon stock per area unit across the Northern peatlands based on random forest model.**

### 3.4.2 Tropical peatlands

In tropical regions, we investigated the main peatlands across three continents: the Peruvian Amazon in Latin America, the Congo basin in Africa, and Indonesian-Papua New Guinea (PNG) in Asia-Pacific. Peat thickness ranged from 0.04 to 10.68

m, with a high variation occurring particularly in the peatlands of Sumatra Island, Indonesia (Fig. 7). Peatlands in Latin America were dominated by shallow peat up to 2 m deep. Several spots in the Amazon Forest were estimated to reach a depth of up to 4 m. In the African continent, the main peatlands in the Congo basin mostly had 1–2 m peat thickness prevalent along the river paths, gradually increasing towards the central peat patches, reaching depths of up to 4 m. In Indonesia and Papua New Guinea, peat varied from 0.5 to 10 m. Shallow peatlands were observed only in the eastern part of Sumatra and coastal

areas in southern Borneo Island.

The carbon stock prediction varied from 50 to 14,000 Mg C ha$^{-1}$, and spatially followed peat thickness prediction (Fig. 7). With a mean (±std. dev) value of 2,327 (±1,395) Mg C ha$^{-1}$ over tropical regions, low carbon stocks were predicted in the shallow peatlands of Sumatra Island, Indonesia. High carbon stocks (> 3000 Mg C ha$^{-1}$) were also found in the Asian-Pacific peatlands, particularly in the deep peatlands of Sumatra and Papua, Indonesia. Peat carbon stocks in Latin America and Africa

were less varied. In the Peruvian Amazon, the peatlands contained approximately 1,038 (±280) Mg C ha$^{-1}$ on average. Congo peatlands had overall higher carbon stocks compared to Amazon, estimated up to 4,400 Mg C ha$^{-1}$, with a mean (±std. dev) value of 1,728 (±484) Mg C ha$^{-1}$.

**Figure 7: Maps of (a) peat thickness and (b) carbon stock per area unit over the main peatlands in Tropical regions based on random forest model.**



### 3.4.3 Southern peatlands

The Southern hemisphere had the least peatlands area, with our prediction indicating predominantly shallow peats (< 0.50 m deep). We highlighted three regions that occupied more than 80% of Southern peatlands, i.e. the southern part of Chile in South America, Tasmania in Australia, and New Zealand (Fig. 8). The South Chilean peatland (1.5 million ha) was predominantly predicted to have a peat layer of less than 1 m. Deeper peatlands (up to 2.10 m) were observed in the Southern Andes and east of the Falkland Island. Tasmanian peatlands (2.05 million ha), occupying 98% of Australian peatlands, were mainly shallow (< 0.5 m peat thickness). The same condition was seen across peat patches in the South Island of New Zealand, yet some were predicted to have thick peat layers (> 2 m deep). In the Kopuatai peat dome, peat thickness increases from west to east, reaching up to 3 m.

Spatial variability of carbon stocks in southern peatlands followed the variation in peat thickness, with deeper peat corresponding to higher carbon stocks (Fig. 8). Overall, the average (±std. dev) peat carbon stock in Southern peatlands was about 703 (±542) Mg C ha$^{-1}$, which is three times lower than tropical peat carbon stocks. High carbon stocks (> 3,500 Mg C ha$^{-1}$) were observed in peat patches of New Zealand. Both Chilean and Tasmanian peatlands mainly (67%) contained carbon stocks of less than 750 Mg C ha$^{-1}$.





**Figure 8: Maps of (a) peat thickness and (b) carbon stock per area unit over the main peatlands in Southern regions based on random forest model.**



## 3.5 Total peatland carbon stock

Table 3 summarises the total carbon stored in peatlands globally and within each of the three latitudinal regions. This table as account for 'peat dominated' locations in the GPM dataset. The total C stock was 1,029 Pg C. Northern peatlands, comprising up to 90% of the global peatland coverage, were estimated to store up to 85% of the total peat carbon stocks (877 Pg C). In contrast, tropical peatlands, occupying only 10% of the global peat coverage, were estimated to hold carbon equivalent to up to 14% of the global value (149 Pg C), with the remaining carbon stock was in the Southern region (3 Pg C).

Assuming that the data are normally distributed, we derived the variation in our peat carbon stock estimation across each region based on the standard deviation of peat carbon density. The global peatland C stock was estimated ranging from 684 to 1,374 Pg C.

**Table 3: Summary of estimated peat carbon pool across the globe and three regions of peatlands.**

| Region | Area (million km²) | Volume (million km³) | Carbon density (Mg C m⁻³) | | Total Carbon stock (Pg C) | |
|---|---|---|---|---|---|---|
| | | | mean | Std.dev | Best estimate | Std.dev* |
| **Global** | 6.57 | 14,610 | 0.070 | 0.024 | 1,029 | 345.1 |
| **Northern** | 5.89 | 12,683 | 0.068 | 0.021 | 877 | 266.3 |
| **Tropical** | 0.64 | 1,449 | 0.108 | 0.036 | 149 | 52.1 |
| **Southern** | 0.04 | 27 | 0.108 | 0.025 | 3 | 0.67 |

*Standard deviation of total carbon stocks was calculated based on the standard deviation of carbon density multiplied by peat volume.

## 4 Discussion

This work demonstrated a digital soil mapping approach to generate global estimates of peat thickness and carbon stocks. We calibrated random forest algorithms fitted to over 25,000 global observations complemented by 19 environmental covariates. Based on current data points, the model performed good estimates of peat thickness, BD, and CC at the global scale, demonstrating the ability to provide accurate predictions across diverse geographic regions.

### 4.1 Appraisal of additional sampled points

In developing thickness models, we incorporated data points randomly sampled from the available peat thickness maps in areas lacking observations. This data augmentation, based on a priori knowledge, successfully constrained the random forest model predictions. Figure 9a shows an example of peat thickness prediction from random forest models that were trained using field observations only, where the thickness was overpredicted to 2-3 m over the Netherlands and Germany. Incorporating 500

points from peat thickness maps of Sweden, the Netherlands, and Denmark (see inset map in Fig.9), resulted in a substantial decrease in peat thickness prediction across European peatlands (Fig.9b). In both the Netherlands and Denmark, the mean peat depth value was decreased by more than 50% (before = 2.3 m, after = 1.08 m), leading to a more realistic prediction with large reductions in peat carbon stock estimation (about 0.5 Pg C in each country) using the same peat bulk density and carbon content

maps.

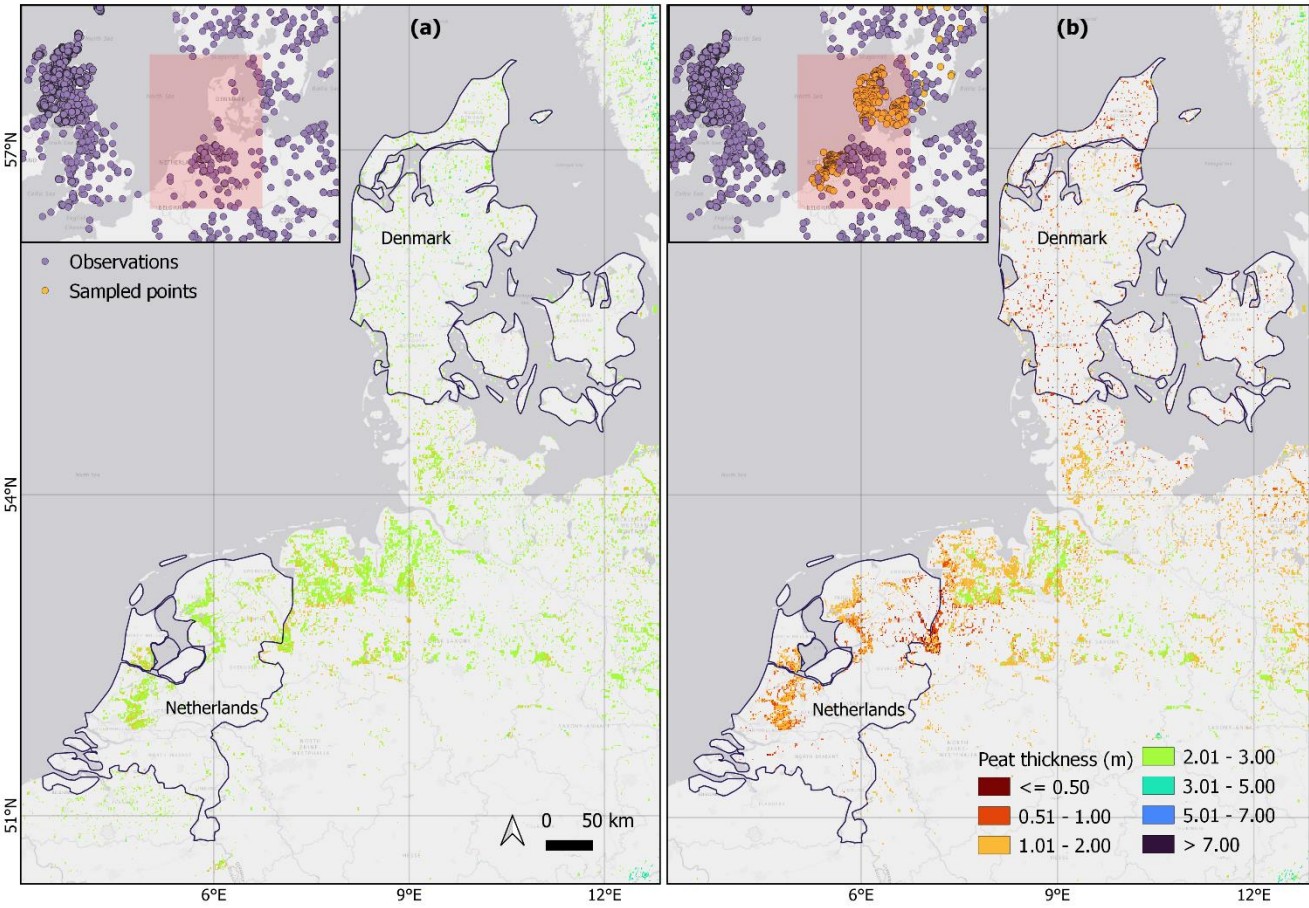

**Figure 9: Comparison of peat thickness prediction in European peatlands (a) without and (b) with additional sampled data points from available peat thickness maps.**

**4.2 Robustness of carbon stock estimation**

We modelled global carbon stocks in peatlands using raster data of predicted peat bulk density and carbon content. This method improves traditional estimation approaches, which typically rely on a single best-value assumption for either peat bulk density or carbon content across a region. Bulk density and carbon content vary spatially due to environmental factors such as topography, climate, vegetation, and human disturbances. We posit that our models effectively captured this variability at a 1 km x 1 km resolution, thereby improving the accuracy of carbon stock estimation.



In our estimation, the global peat carbon stock is 1,029 Pg which is much higher than previous studies, which reported values ranging from 445 to 612 Pg C (Table 4). This is primarily due to the larger peat extent based on the UNEP Global Peatland Map.

**Table 4: Comparison of global peat carbon stock estimations from previous studies and this study.**

| References | Peat Area (million km²) | Total carbon stock (Pg C) | Notes |
|---|---|---|---|
| Joosten (2009) | 3.81 | 445.6 | Average peat depth of 2 m. |
| Immirzi et al. (1992) | 3.97 | 329-528 | Calculated based on 1.5-m peat depth |
| Page et al. (2011) | 3.99 | 479.7 | Revising tropical area of the Immirzi's estimation; calculated based on 1.5-m peat depth of global average. Bulk density = 0.09 Mg m$^{-3}$; Carbon content = 0.56 g g$^{-1}$. |
| | 3.99 | 610.0 | Revising tropical area of the Immirzi's estimation; calculated based on 2.3-m peat depth, average value for Northern peatlands. Bulk density = 0.09 Mg m$^{-3}$; Carbon content = 0.56 g g$^{-1}$. |
| Yu et al. (2010) | 4.41 | 612 | Carbon dating and modelling approach |
| Leifeld and Menichetti (2018) | 4.63 | 597.8 | Map compilation and statistical average |
| This study | 6.57 | 1,029 | Digital soil mapping approach |


Our raster data of peat extent, provided by the Global Peatland Initiative, covers an area of 6.57 million km² designated as 'peat dominated' lands, which we assumed to be peatlands. This number exceeds the estimates reported in the GPA 2020, which accounts for up to 4.8 million km² of peatlands, excluding any peat dominated area with less than 30-40 cm peat layer. This means our estimation includes 1.77 million km² of peatlands that were previously classified as non-peats with less than

30-40 cm peat layer. However, it is important to include as much of the probable known peatlands as possible to comprehensively estimate their carbon stock.

In particular, the Indonesian peat extent (20.1 million ha) that we used to update the GPM resulted in estimates of carbon storage, approximately 65.9 Pg. This number doubled the estimates from a previous study using the Indonesian peat map from Wetlands International (30.79 Pg C, area=20.9 million ha) (Warren et al., 2017). This high estimation was attributed to the





higher carbon density (mean = 0.098 (±0.029) Mg C m$^{-3}$) and thickness compared to the value used in Warren et al. (2017) for total peat stock calculation (0.055-0.069 Mg C m$^{-3}$).

## 4.3 Comparison of the global maps to regional predictions

We compared PEATGRIDS to previous studies that utilised machine learning models to generate maps of peat thickness and carbon stocks at the regional scale. We noted that the resolution of PEATGRIDS is limited to 1 km grid cell area. These

comparisons primarily focus on the visual analysis of prediction maps and peat thickness estimates. In addition, the estimated peat carbon stock was also compared, highlighting differences in the spatial extent of peat areas.

First, PEATGRIDS were compared to maps of peat thickness and carbon stocks across the Congo basin in Africa by Crezee et al. (2022). Their mapping effort utilised RF algorithms to model peat thickness based on four environmental predictors, including the distance from the peat margin. Carbon stock was derived from the linear relationship between peat carbon stocks

and the square root of peat thickness. Generally, PEATGRIDS showed a similar pattern with the maps of Creeze et al. despite less distinct changes in value between river paths and peat dome areas.  For peat thickness, PEATGRIDS retrieved similar mean values, which is about 1.7 m. Both maps had similar thickness variations, with the maximum class being less than 5 meters thick. Moreover, PEATGRIDS  peat thickness and carbon stock showed similar spatial variability to the previous study (Fig. 10). Surprisingly, despite using a different method to estimate carbon stock, the PEATGRIDS estimation of total carbon

stock over the Congo Basin was relatively similar to previous studies. With a 13% smaller peat area (PEATGRIDS compared to previous studies), PEATGRIDS estimated 15% less peat carbon stock (25.4 Pg) compared to the previous estimate of 30 Pg C (Crezee et al., 2022; Dargie et al., 2017).

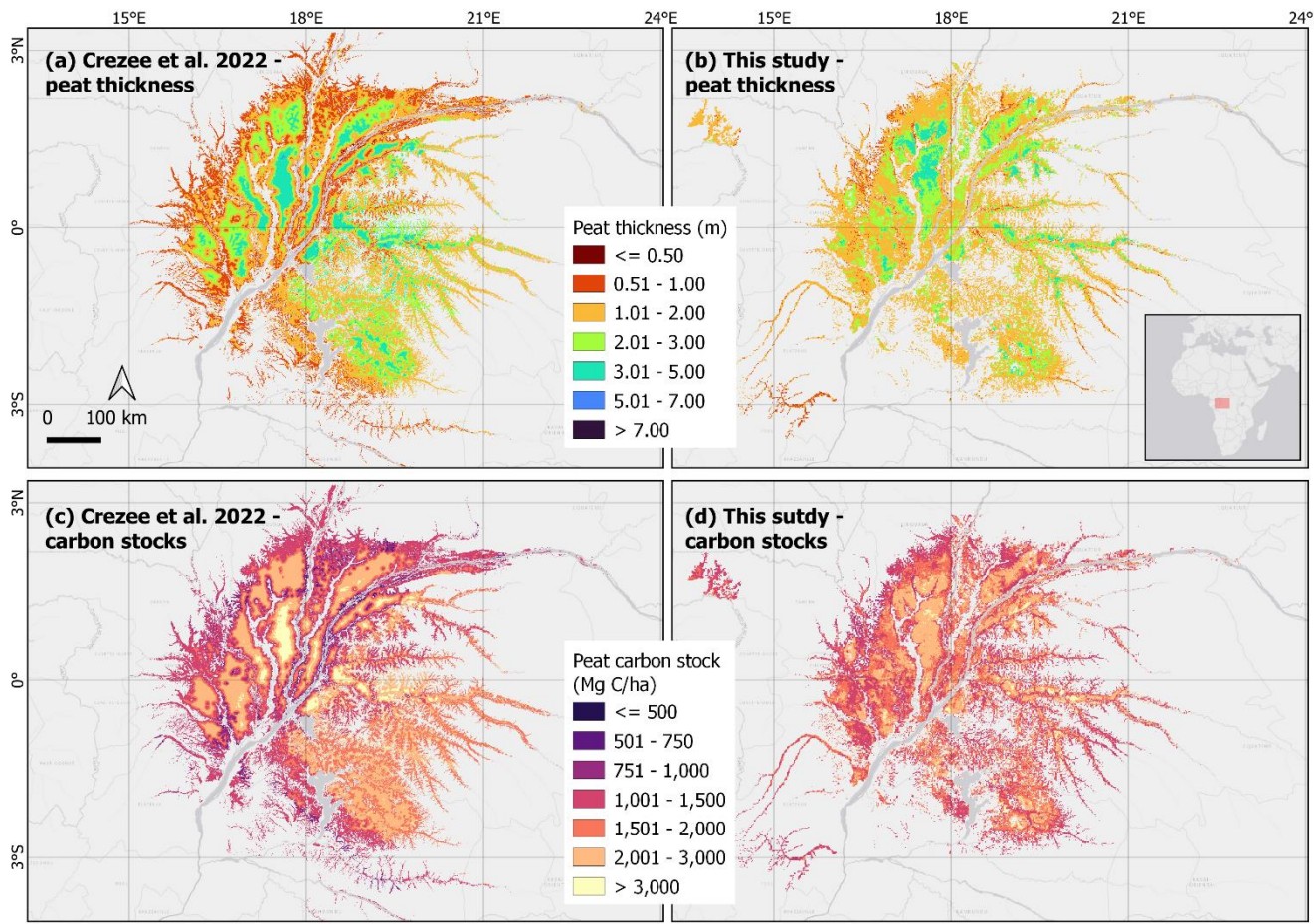

**Figure 10: Comparison between the modeled peat thickness and carbon stocks in Congo basin peatlands from (a; c) the previous study by Crezee et al. (2022) and (b; d) this study.**

In another tropical area, the Peruvian peatlands, PEATGRIDS thickness map was compared to the 100 m resolution map generated by Hastie et al. (2022) (Fig. 11). Their peat thickness distribution map was derived using RF algorithm trained on 1,359 data points according to remote sensing layers combined with distance to peatland edge and height above nearest data. Their peat carbon stock was estimated according to a linear relationship between peat carbon stock and peat thickness. In this peatlands, PEATGRIDS covered less area (53,000 km$^2$) than Hastie et al. 2022 (62,700 km$^2$). Overall, both thickness predictions covered the same range of values but exhibited different spatial variability. Despite using a similar algorithm, the difference likely arose from the inclusion of the distance to the peatland edge as a model input by Hastie et al., which was not included in the PEATGRIDS models. PEATGRIDS predicted deep peatlands (> 3 m) covering only the eastern part of the Peruvian Basin, leading to a lower overall peat thickness (mean = 1.58 m) than reported by Hastie et al. (2022), which is around 2 m. Regarding total carbon stock, PEATGRIDS derived slightly higher value (5.54 Pg) than their estimates (5.38 Pg).





Additionally, a comparison between PEATGRIDS thickness and observed data in peat patches across the upper Ucayali Valley in Peru, as reported by Crnobrna et al. (2024), showed a similar range (PEATGRIDS = 1.31-1.70 m, observed data = 1.36-2.30 m). This evidence demonstrates that PEATGRIDS can accurately capture the thickness variation of small peatland patches.

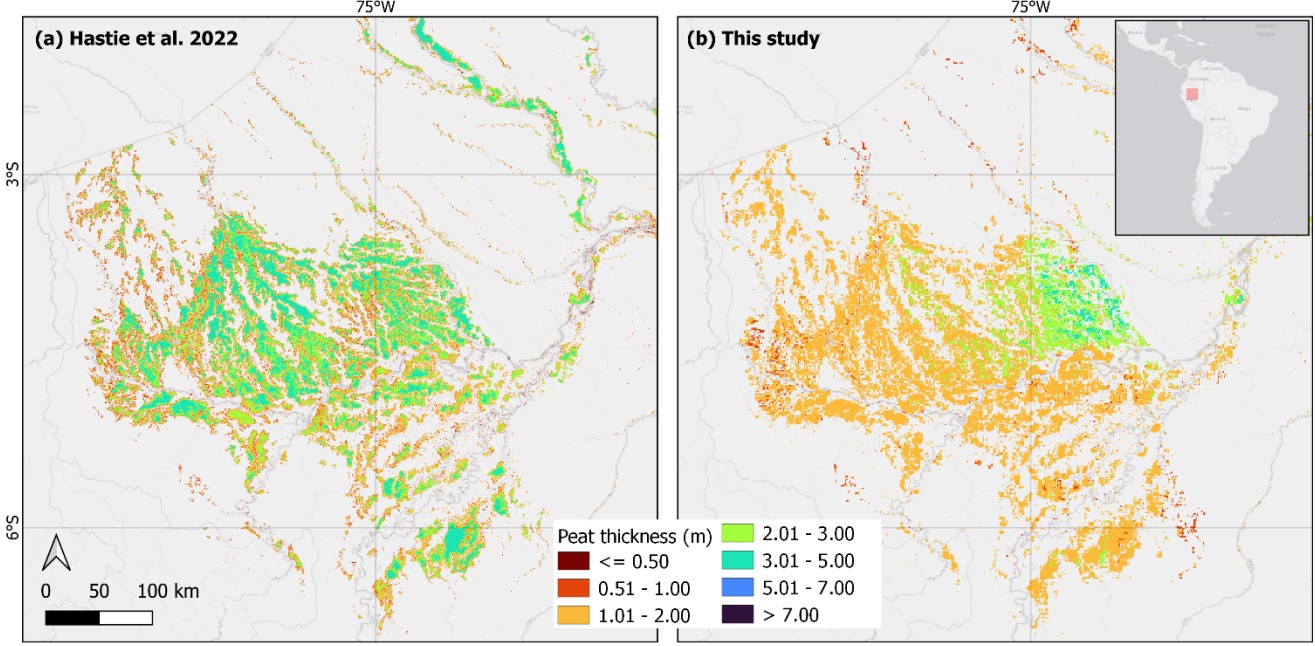

**Figure 11: Comparison between the modelled peat thickness in Peruvian basin peatlands from (a) the previous study by Hastie et al. (2022) and (b) this study.**

PEATGRIDS were also compared to the estimates of a part of peatland in Bengkalis island, Indonesia (approximately 50,000 ha) by Rudiyanto et al. (2018). Their thickness map was derived from the cubist tree models, while the carbon stock estimation was by multiplying the peat volume across peat thickness by the average carbon peat density from field data and literature (0.085 Mg C m$^{-3}$). Despite the coarser spatial resolution, PEATGRIDS had similar spatial variability, showing deeper peat layers (more than 7 m) in the eastern mid-part of the island. This agreement was further supported by the pattern of the carbon stock map. PEATGRIDS estimates on total carbon stock across the same extent resulted in a similar number, being about 250 Megatonnes C, to the Rudiyanto's estimation (252 ± 76 Megatonnes C).

In Scotland, peat thickness and carbon stock mapping have been performed at 100 m resolution using the neural networks model by Aitkenhead and Coull (2019). Their mapping effort used 10,141 data points for the model training and focused on estimating peat carbon content and bulk density at 5 cm increment up to 10 m depth. Their peat layer was defined by soil layer containing carbon more than 0.2 g g$^{-1}$, and peatland was defined as an area with a peat layer of more than 50 cm. The GPM extent over Scotland overly predicted area along the coastal line as peatlands, leading to higher peat extent (the GPM = 31,761





km², Aitkenhead and Coull = 23, 958 km²). Generally, PEATGRIDS exhibited deeper peat layer across Scotland, around 1-2 m thick. This was likely due to generalisation by RF algorithm. This led to a higher estimation of carbon stock per unit area and total carbon stock across Scotland. PEATGRIDS carbon stock prediction had a wider range of carbon stock (234-7,360 Mg C ha$^{-1}$) compared to Aitkenhead and Coull (2019), who estimated the stock to be between 101-6,300 Mg C ha$^{-1}$.


On a larger scale, we compared PEATGRIDS to the mapping effort in Northern peatlands (Hugelius et al., 2020). Their thickness map was derived from RF models trained on 7000 data points, covering an area of 3.7 million km², excluding area with less than 40 cm peat thickness. To calculate carbon stocks, they used peat thickness to estimate peat organic carbon using linear relationships formulated based on the peat core data. Statistically, PEATGRIDS estimation on the peat thickness across Northern peatlands was slightly lower (mean=2.15 m) than Hugelius et al.'s prediction (mean=2.49 m). In addition, we found that PEATGRIDS carbon stock (mean = 1,445 Mg C ha$^{-1}$) was in the range of their prediction (1,150 ± 41 Mg C ha$^{-1}$). Since PEATGRIDS may include areas with a peat layer of less than 40 cm but exclude permafrost areas across northern Canada, comparing the estimates of total carbon stock with the previous study is challenging.


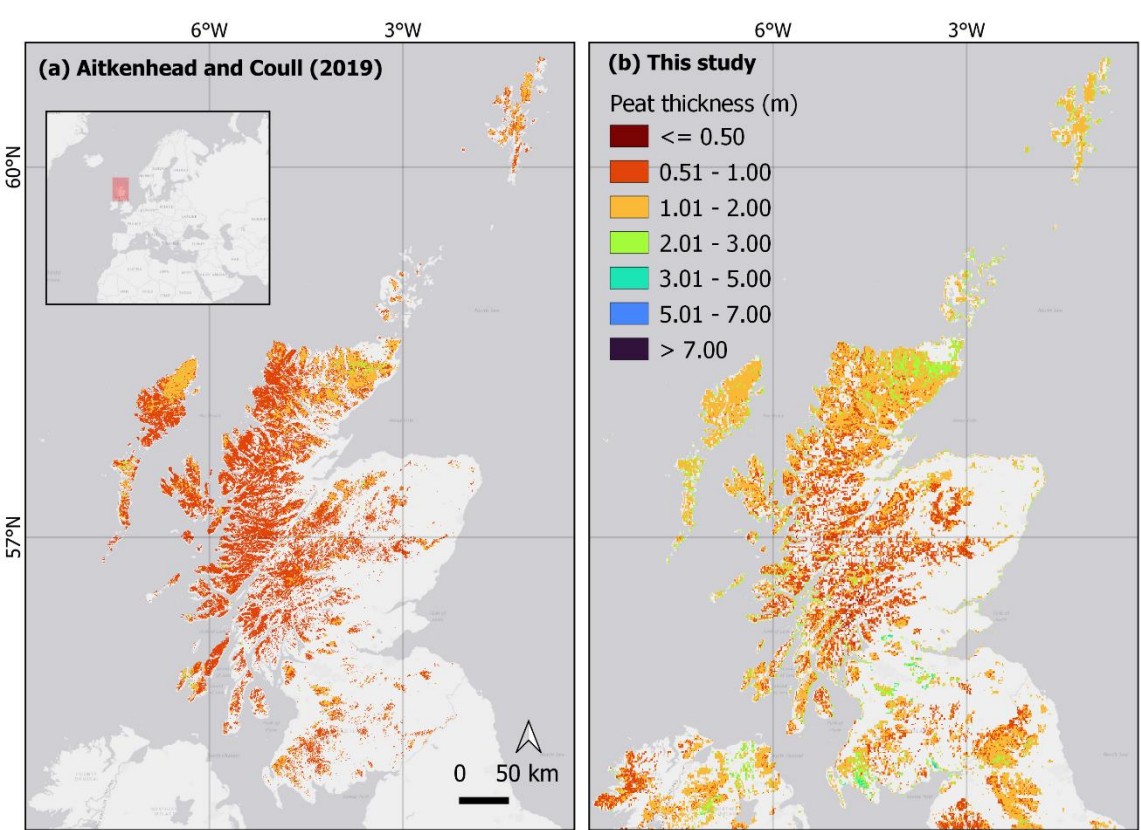


**Figure 12: Comparison between modelled peat thickness in Scotland from (a) a previous study by Aitkenhead and Coull (2019) and (b) this study.**



The estimate of global soil organic carbon (SOC) stock (up to 2 m) is around 2,400 Pg C (Batjes, 1996), with the mean carbon
stock for Histosols being 2,180 Mg C ha$^{-1}$. More recently, the global SOC stock has been updated to 3,000 Pg C, comprising
all depths of soil layers, yet there is large uncertainty, particularly for depths >1 m (Köchy et al., 2015). PEATGRIDS estimate
(1,029 Pg C over 6.57 million km$^2$) suggests that peatlands, which occupy 5% of the land area, constitute more than a third of
the global carbon stock. This indicates that the global SOC stock may be underestimated. Therefore, further work is needed to
incorporate peatland data into the global SOC map to represent global carbon stock accurately.

**4.4 Model limitations and possible improvements**

While PEATGRIDS have mapped peatland thickness and carbon stock across the world, we recognise some limitations that
need further improvement. The peat extent used in this study is based on the Global Peatland Map, which may include areas
not recognised as peats in different classification systems, or areas that have undergone significant land use changes. Since no
universal definition of peat exists, PEATGRIDS provides the first estimate for global peat-dominated areas. The 1 km spatial
resolution may overestimate some peatlands, especially those that cover areas less than 100 ha, or overlook smaller peatlands.
Future refinement of the global peatland extent may improve the accuracy of the peat extent map.

The availability of observation data limits the accuracy of the models. Regarding peat thickness prediction, observation data
in the tropics and southern regions is still limited. For example, in Africa, 174 data points, specifically measured in the main
peatlands of the Congo basin, were used to train the random forest model for peat thickness across 0.3 million km$^2$ of peatlands.
This means the density of available data was very low (< 0.001 km$^{-2}$), which might limit the models' ability to capture the
variability of peat thickness and carbon stock outside the basin. We attempted to fill the data gaps with available peatland
maps, but information on peatlands in many regions, including Africa, remains scarce.

Similarly, in predicting bulk density and carbon content, our dataset was dominated by data points from Northern peatlands,
particularly those in West Siberia. This prevalence may have influenced the random forest models, causing them to estimate
values for regions without data based on the mean of the observed values, most likely leading to overestimation. Therefore,
incorporating additional observation data from less-sampled regions would enhance the reliability of the models.

To estimate the multivariate peat parameters, this work focused solely on testing the random forest algorithm. Despite its
popularity for delivering high accuracy and robust results, the RF algorithm does not always perform best (Rudiyanto et al.,
2018). Alternative ML algorithms or advanced techniques, such as deep learning, could potentially enhance model accuracy
when predicting multivariate variables.

One important information not addressed in this work is the uncertainty of the predicted maps. Uncertainty analysis is necessary
to evaluate how reliable the predicted maps are for decision-making processes, as it acknowledges model limitations and
interpretability (Wadoux et al., 2020). Model validation metrics can be used in the interim as an indication of reliability.

Nevertheless, we have generated PEATGRIDS as the first global peat thickness and carbon stock maps, and the workflow is
documented in an open-source environment. This allows for easy updates to PEATGRIDS data product as new data becomes

available. With further improvements and more observation data, we could provide a more accurate picture of the global peatland carbon stock.

**Data Availability**

PEATGRIDS version 1.0 is available in Zenodo https://doi.org/10.5281/zenodo.12559239 (Widyastuti et al., 2024).

**Conclusions**

This work introduces PEATGRIDS, a data product of peat layer thickness and its carbon stock estimation at the global scale, providing baseline for peatlands monitoring. Using topography, climate, and land surface covariates, random forest models were used to estimates peat thickness, multi-depth bulk density, and carbon content. PEATGRIDS revealed that the current global 'peat dominated' soils store carbon up to 1,029 Pg C, with the Northern region holding more than 85% of the total. This

product represents an initial step in mapping the condition of global peatlands to enhance our understanding of these ecosystems. The modelling framework is open source, allowing for easy updates to the map as new data becomes available. We anticipate this work will encourage the collection of new data points and the refinement of modelling strategies, which will improve predictive accuracy over time. In addition, this study suggests that the global SOC stock could be underestimated and requires an update to include these peatlands data.

**Author Contribution**

All authors contributed to the scientific research and writing of this paper. MTW: Designing computer programmes, Writing - Original Draft, Formal analysis, Visualization. BM, JP and FM: Conceptualization, Data Curation, Writing - Review & Editing. MA, AB, JC, DF, DK, YM, FMc, CR, R, BIS, and MT: Resources, Writing – review & editing.

**Competing Interests**

The contact author has declared that none of the authors has any competing interests.

**Acknowledgements**

This research was supported by ARC Discovery Project Forecasting Soil Conditions DP200102542. MW was funded by Lembaga Pengelola Dana Pendidikan (LPDP) Scholarship (LOG-7157/LPDP/LPDP.3/2023). We would like to thank R. Deragon for sharing data from peatlands in Eastern Canada. We acknowledge Justin Wyatt and his colleagues at Waikato





Regional Council for compiling and sharing the peatland data of New Zealand. We thanks Louis Gillet for providing data of peatlands in Ireland, and Christoph Kratz at Nature England for providing peat data in England.

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
