# Peer review of "PEATGRIDS: Mapping thickness and carbon stock of global peatlands via digital soil mapping"

_Earth System Science Data, 2024_

## Author Comment (AC1)

**Reviewer 1**

This manuscript aims to address the uncertainty in global peatland extent and peat carbon stocks by developing a global model of peat thickness and carbon density. The method that is applied and the resulting maps will be useful to peatland ecologists and soil scientists around the globe, and highlight the important role that peatlands play as carbon reservoirs.

Interestingly, the authors' overall conclusion that peat soils globally store more than 1,000 Pg C is much larger than previous estimates. However, given the discoveries of new peatlands in recent years, particularly in the tropics, which indicates a past general trend to underestimate peatlands, this much larger estimate could well be more accurate. This manuscript will therefore make a valuable contribution to a growing body of literature that tries to pinpoint and understand the stocks and flows of peat carbon in the face of global environmental change.

*General comments:*

Overall, the application of digital soil mapping to estimate global carbon stocks is a useful approach that hasn't been tried before on this scale, as far as I know. The authors have collected an impressive dataset of peat thickness, BD and CC data and used this to train random forest models for six regions. This methodology appears sound and well applied, given the obvious limitations to modelling on such a large scale.

In terms of outcomes, it seems that the authors' mean values correspond well with previous work, particularly in the 5 case studies, but that their model is often struggling to capture the regional variability that most of these smaller-scale maps do show. Essentially, the authors' model is moving closer towards the average at the expense of regional variation. This is understandable and as expected, given the use of a RF model with sometimes limited training data from certain regions. However, this also means that it would be good to stress the global nature of this map. The results should be treated with more caution at regional scales by end-users, especially if more local maps are also available.

We thank you for your positive feedback and careful review. We have added the caveats and stated the nature of the global map.

Additionally, I had a couple of general questions:

- It was unclear to me whether some of the training data taken from external sources includes modelled data itself. Could you clarify if all of the input data are direct field measurements, or whether this includes modelled data from previously created maps? In the latter case, it would be good to make this more explicit (how much true field data, how much modelled? In which areas?) For example in Table S1. Also, what does this mean in terms of error propagation from previous sources into this new model? How does this affect your model's uncertainty?

    Thank you very much for your questions. Data points for model training are derived mainly from field measurements and complemented with data from peat maps. The maps data are spatially estimate of peat thickness (i.e. Denmark and Netherlands), and regional peat extent maps according to peat depth classification (i.e. Indonesia and Sweden). We have clarified the number of measured and extracted data along with its area in Table S2 and S3. Since the maps were the only data in the areas, we treated them as observation, recognising the uncertainty that comes along with the maps. As we

demonstrated in the paper, adding these "maps" data, produced much more realistic peat thickness and more accurate European peatlands and not significantly affect the model's uncertainty.

- It is not entirely clear to me based on what criteria the predictor variables were chosen. For example, although the list of predictor variables includes a wetness indicator such as the Topographic Wetness Index, I was left wondering whether it could have been useful to explicitly add a variable related to the seasonality of wetness/inundation? Some peatland areas, particularly along river valley systems in the tropics, might experience seasonal droughts that could be a driver of decomposition, and therefore influence thickness, BD and/or CC. (Conversely, they might experience extreme flooding during the rain season as well). Such seasonal variability in wetness/inundation is currently not captured by your list of predictor variables, as both your WordClim and PALSAR variables use yearly averages only. Perhaps it could be useful to include precipitation seasonality, precipitation of the driest month, or a similar variable from WorldClim?

Thank you very much for your valuable insights. We have clarified about how we chose the model predictors, and the relevance of each variable towards the presence of peatlands. Regarding seasonal variable, we haven't covered this in our current model, as the current global map is just a snapshot based on existing data collected over years.

The model uses six geographic regions, at least for training BD and CC models. This seems a logical methodological choice. However, I was wondering if it would be useful to apply a spatial cross-validation approach as well, in addition to the five-fold CV currently used during training. Currently, all training and testing data are randomly taken from the same region, which means that they could well be close to each other and show spatial autocorrelation. To account for this and test the model's accuracy in areas from which it lacks any training data, it would be good to predict a test area that has not been used for training. For example, by using five regions as training data and testing on the sixth one. Or by setting testing blocks apart within each of the six continental regions. This way, the authors would get a better idea of their model's accuracy, given some regions have very limited data.

The issue of spatial autocorrelation has not been addressed yet at this stage. Having multi models to predict one parameter globally can be exhausting, as we need more computational resources to do it. We have limited resources to generate the maps globally within short period. Spatial cross validation is not appropriate in this case as the region is too large to make any meaningful extrapolation.

*Detailed comments:*

Line 23/27: Harmonize and be consistent in number of datapoints that has been used. Currently, the different numbers (25,000/25,200) are confusing. If the numbers differ for thickness, BD and CC, give the numbers for each of these datasets explicitly in the Methods.

We have clarified the number of data points for each variable.

Line 47-50: Harmonize the use of million hectares and million km2. Choose one or the other, but not both interchangeably.

We have harmonised the area unit into million km2 throughout the text.

Line 61: This line should state that *peatland carbon stocks* were mapped by this paper using a random forest model, not peatland extent (as is currently implied).

We have revised this as suggested.

Line 81: This says that the GPM reports 8.7 million km2 of peatland (of which 6.7 million is peat-dominated). However, on line 50 you state that the GPM estimates peatlands to be 4.9 million km2. This appears to be a contradiction. What is the correct number?

In revised MS, we clarified that the peat map we used is from the Global Peatland Database, and hereafter referred as the GPM. The GPM covers 9.03 million km2, then we updated it by adjusting peatlands coverage in Indonesia. The global extent reduces to 8.84 million km2. This area still consists of two classes of peatlands: peat dominated and peat in soil mosaics. We use this for further analysis.

Line 82: What are the other non-peat dominated lands that you have excluded from the GPM? Could you elaborate why these are not useful in this study? In general, it would be good to remind the reader here that the GPM has no specific peat definition globally. As your final carbon stock number depends a lot on the GPM's area estimate, it would be good to say something about how this could impact your results.

Since we could not find the specific explanation about the definition of non-peat dominated area, we include that class in our revised MS.

Line 97-98: This additional map that was used to extract more points from Indonesia does not appear to be in Table S1? Please clarify.

We have mentioned the map in the MS. However, in the revised text, we have added it into the list in Table S1.

Line 112: What did you do if only carbon density was provided in certain datasets, and not the underlying BD and CC measurements? How were these datapoints included in your models for BD, CC and the final carbon density output? For example, the Congo Basin is one of your case studies, but you did not include any BD (and only 1 CC) values from Congo in the training set. However, carbon density values are publicly available from this area.

We did not include them into any models for BD and CC. Instead, we used them to verify our carbon density maps that we derived from multiplying BD and CC maps.

Line 309: Please look at sentence 'This table as..' Does not read very well.

This line has been revised accordingly.

Line 370: Creeze et al. (2022) is misspelled

This line has been revised accordingly.

Line 371-374: Please specify the full mean and SD values of the previous study as well as your study for comparison. You say they are similar but do not give the regional values to back this up. This also applies to the other case studies: sometimes this data is provided, but not in all

cases. It would be useful to add a Table that compares the mean thickness and carbon density values, and total carbon stock, for the 5 case studies from Congo, Amazon, Indonesia, Scotland, and Northern peatlands, for both your map and the original studies.

We have clarified this section. We only refer to the results in Congo basin to studies by Crezee et al and Dargie et al., of which we mentioned the number of their estimations in the last sentence. Thank you for your suggestion regarding the table.

Line 443: 'specifically measured in the main peatlands of the Congo Basin' Are these selected from this paper's peat thickness map (which is modelled), or are these original field-measurements taken from the same study? (see general comment above)

This refers to measurement data that available from the same study. We did not use any peat maps for Congo Basin to get additional data points.

Line 446: 'We attempted to fill the data gaps with available peatland maps, but information on peatlands in many regions, including Africa, remains scarce.' So did you do this or not? Table S1 lists only point data in the tropics, which implies that you did not use additional map data? Please be more explicit here.

For Africa, we did not extract additional data points from the modelled peat maps because we already have the measurement data of it, which is the same as they used to build that map. We refer to conventional map that does not exist in Africa. We have revised this sentence by mentioning what kind of map that we refer here.

Line 456: Why is uncertainty not addressed? This should be relatively straightforward to assess with a RF model in GEE.

We have revised the model and trained QRF models in Python environments as it provides the function to fine tune RF hyperparameters. In the revised MS, we provided the uncertainty for all predicted maps of peat thickness, BD, CC, and carbon stocks.

Section 4.4: I would appreciate a line in here that stresses that your work and the resulting maps are useful in estimating global carbon stocks, but that at regional scale your model appears to be not fine-grained enough to capture most of the known regional variations in the 5 case studies. Hence, it should be used with caution for assessing regional peatlands.

Thank you for your valuable insights. The suggestions have been added.

Supplement Table 1: Please list the number of samples that you used from each study, and (if necessary) specify which of these sources provided direct field observations, and which of these sources provided modelled thickness/BD/CC values. This seems an important distinction to make.

The additional sampled data was only for peat thickness. We did not use any peat BD and CC data extracted from maps. We have added Table S2 as a complement to Table S1 with the number of data from field measurements, extracted from maps, the total after we combined them and after we aggregated them. We also specified the data based on the specified area and the references.

**Reviewer 2**

The study attempts to map peat thickness and carbon stock in peatlands across the world using 'observations' (some are truly observations, others are from existing regional maps/models) to train and test random forest models, extrapolated using remote sensing and geodata products. Their models are spatially constrained by the existing Global Peatland Map (GPM, restricted to the 'peat dominated' lands = 6.7 million km$^2$), and further adjusted in Indonesia (based on Haryono et al., 2011) (= 6.57 million km$^2$).

Despite the conservative decision to restrict their predictions to 'peat dominated lands', this study predicts a total global peat carbon stock of 1,029 Pg C, which is between 1.7 and 2.3-times previous estimates.

If valid, this is a bold and significant conclusion, and the dataset could be useful for wider research and policy communities. However, there are several substantial issues with the methodology which I believe need to be resolved before the paper could potentially be accepted.

Thank you for your positive feedback on our work.

**Major comments**

- Lack of uncertainty assessment and carbon density prediction:

I appreciate that you are upfront about the lack of uncertainty assessment, but I don't think this is acceptable in the context of a global assessment of the peat carbon stock, especially when your total carbon stock value is so large, while also considering other methodological limitations (to follow).

Linked to this is the performance of your carbon density prediction (Figure 5). Not only does the model perform poorly against observed carbon density measurements but it appears to be systematically biased- overpredicting carbon density. You do initially acknowledge this in line 250. However, you make no reference to it in the discussion where you conclude (line 345) – 'In our estimation, the global peat carbon stock is 1,029 Pg which is much higher than previous studies, which reported values ranging from 445 to 612 Pg C (Table 4). This is primarily due to the larger peat extent based on the UNEP Global Peatland Map.'

Figure 5 contradicts this conclusion. While the larger peat extent certainly accounts for some of the higher carbon stock, your overprediction of carbon density is a significant factor which needs to not only be discussed but quantified (Figure 5 suggests that it may account for as much as 50% of the increased carbon stock estimate).

For examples of estimating peat-carbon stock uncertainty (and propagating uncertainty in underlying variables such as peat thickness), see Hugelius et al. (2020), Crezee et al. (2022), Hastie et al. (2022), Draper et al. (2014) etc.

You also have a large RMSE for your peat depth thickness prediction in some regions (e.g. North America and SE Asia), (Table 2).

Figure 5 was used to extend the validation of our peat BD and CC predictions. In this case, we took advantage of carbon density data that are available for some peat regions, like in Africa. The contradictive statements have been revised accordingly.

In the revised MS, we provide the updated predictions of peat thickness, BD and CC since we have additional data from peatlands in Canada. Our revised manuscript and data also provide uncertainties calculated using the Quantile RF models. We also calculated the final uncertainty maps for the global map of peat carbon stocks by propagating the standard deviation of each map of peat thickness, multi-layers of BD and CC.

- Definition of peat:

In line 45 you write- 'The recent global peatlands assessment (GPA) by United Nations Environment Program (UNEP) reported an updated global peat coverage, reaching up to 500 million ha by defining peatlands as areas with more than 30 cm of peat layer (UNEP, 2022)'

You are also using the GPM to constrain peatland area. As such I assume that you are using this 30 cm cut off for your peat definition.

However, later in line 274 you write- 'Peat thickness ranged from 0.04 to 10.68 m, with a high variation occurring particularly in the peatlands of Sumatra Island, Indonesia (Fig. 7)'

0.04 m or 4 cm does not qualify as peat under the GPA definition. Do you therefore exclude areas which are predicted to be < 30 cm (based on your model) from your results, and carbon estimation? The above sentence suggests not.

Thank you for your corrections. In the revised MS, we clarify the correct number of peat extent that we used for analysis. The original peat extent from Global Peatland Database is 9.03 million km-sq, and then we adjust the peatlands in Indonesia, resulting in 8.84 million km-sq. The exclusion of peat thickness <30cm, and non-peat dominated class was done in our revised analysis.

- Covariates and parameters:

In line 138 you write- 'We used 19 covariates (Table 1) representing peat formation factors to predict peat thickness, BD, and CC separately.'

How did you test for redundancy of driver variables (e.g. cross correlation) and model overfitting? Please better explain and justify your model set-up (in particular the selection and retention of driver variables).

The variables selection was done by theoretical relevance of each predictor towards the peat forming factors of which the global rasters data are available at high resolution. We addressed model overfitting issue by finetuning the QRF hyperparameters separately for each model. As we have limited number of covariates, we did not find redundant or significant correlation of predictors.

- Spatial autocorrelation:

Your modelling scheme does not seem to account for spatial autocorrelation. As an additional step, you could for example employ a spatial cross validation approach to get a better understanding of model performance. At the very least you should discuss the issue of spatial autocorrelation and potential implications. See for example- Garcia, M (2021); Meyer H and Pebesma (2022), Golblatt et al (2016).

It is correct that we did not address the issue of spatial autocorrelation. We calibrated and evaluated the model via 6 regions. A discussion regarding to this matter has been added in the revised paper. In addition, the topic of spatial cross validation is also being discussed in

Wadoux et al. 2021 and De Bruin et al. 2022. Spatial cross validation may not produce a better understanding of model performance due to the nature of legacy soil data collected from multiple sources.

Wadoux, A.M.C., Heuvelink, G.B., De Bruin, S. and Brus, D.J., 2021. Spatial cross-validation is not the right way to evaluate map accuracy. Ecological Modelling, 457, p.109692.

De Bruin, S., Brus, D.J., Heuvelink, G.B., van Ebbenhorst Tengbergen, T. and Wadoux, A.M.C., 2022. Dealing with clustered samples for assessing map accuracy by cross-validation. Ecological Informatics, 69, p.101665.

**Specific comments:**

In line 49 you write- 'According to the GPM, the global peatland area reached 4.9 million km2..'.

In line 81 you write- 'The GPM, available at 1-km resolution, reports up to 8.7 million km2, double the the peat area…'.

These two sentences (above) seem contradictory, please change or explain the discrepancy (e.g. different version of map or peat definition?).

We used the GPM provided by the Global Peatlands Database via this source: https://greifswaldmoor.de/global-peatland-database-en.html . According to the provider, this map data is the base map of the GPM v2.0 as reported by UNEP in Global Peatland Assessment (GPA) 2022 (reported to have coverage 4.9 million $km^2$). Since the map mentioned in the GPA is not published yet, we use this GPM as peat extent in our modelling effort.

Line 140-'We selected the hyperparameter values with the highest cross-validation score as the final model.'

I see from Table S2 that you tested hyperparameters within a defined range. It would be good to explain why you chose these ranges in terms of e.g. avoiding overfitting, as from the main text it seems that you chose the hyperparameters only based on model performance.

Correct. We performed finetuning the QRF hyperparameters to avoid overfitting on the final models. The chosen hyperparameters are those that highly affect the formation of trees and its randomness. Values to be assigned for each hyperparameter depend on its close variation from the default value of RF function in Python environments with regards to its min or max value.

In line 345 you write- 'In our estimation, the global peat carbon stock is 1,029 Pg which is much higher than previous studies, which reported values ranging from 445 to 612 Pg C (Table 4). This is primarily due to the larger peat extent based on the UNEP Global Peatland Map.'

See previous comment, what about bias in carbon density prediction?

We have revised this according to the updated results on carbon density prediction and its uncertainty.

In line 350 you write- 'Our raster data of peat extent, provided by the Global Peatland Initiative, covers an area of 6.57 million $km^2$ designated as 'peat dominated' lands, which we assumed to be peatlands. This number exceeds the estimates reported in the GPA 2020, which accounts for

up to 4.8 million km$^2$ of peatlands, excluding any peat dominated area with less than 30-40 cm peat layer. This means our estimation includes 1.77 million km2 of peatlands that were previously classified as non-peats with less than 30-40 cm peat layer. However, it is important to include as much of the probable known peatlands as possible to comprehensively estimate their carbon stock.'

Related to above comment, are you also excluding 'peat' pixels where your model predicts <30cm of organic soil (peat) thickness?

No. For the final product, we did not exclude peat thickness < 30 cm. But we also provided scenarios on the extent and C stock.

In line 383 you write- 'Their peat thickness distribution map was derived using RF algorithm trained on 1,359 data points according to remote sensing layers combined with distance to peatland edge and height above nearest data.'

Do you mean '…height above nearest drainage.'?

Yes. Thank you for your correction.

Line 435- '4.4 Model limitations and possible improvements While PEATGRIDS have mapped peatland thickness and carbon stock across the world, we recognise some limitations that need further improvement. The peat extent used in this study is based on the Global Peatland Map, which may include areas not recognised as peats in different classification systems, or areas that have undergone significant land use changes. Since no universal definition of peat exists, PEATGRIDS provides the first estimate for global peat-dominated areas. The 1 km spatial  resolution may overestimate some peatlands, especially those that cover areas less than 100 ha, or overlook smaller peatlands. Future refinement of the global peatland extent may improve the accuracy of the peat extent map.'

I would suggest also mentioning that restricting the study to the GPM definition of 'peat dominated' areas could also result in missing some peatlands, such as over Brazil (see for example Hastie et al., 2024 and Gumbricht et al., 2017).

In the updated version, we provided the predicted map covering all areas of the GPM map.

Line 456- 'One important information not addressed in this work is the uncertainty of the predicted maps. Uncertainty analysis is necessary to evaluate how reliable the predicted maps are for decision-making processes, as it acknowledges model limitations and interpretability (Wadoux et al., 2020). Model validation metrics can be used in the interim as an indication of reliability. '

Considering the poor performance of your carbon density model and bold conclusions (i.e. 1,029 Pg C), an assessment of uncertainty is essential.

We revised the models and provide the uncertainty maps in the revised version. Along with this, we revised all prediction numbers and its discussion accordingly.

**Additional references mentioned-**

Draper *et al* 2014 *Environ. Res. Lett.* **9** 124017- https://iopscience.iop.org/article/10.1088/1748-9326/9/12/124017

Garcia M 2021 *Investigating the Use of Spatially-Explicit Modelling and Cross-Validation Strategies in Spatial Interpolation Machine Learning Problems* available at: https://run.unl.pt/handle/10362/113881

Goldblatt R, You W, Hanson G and Khandelwal A K 2016 Detecting the boundaries of urban areas in india: a dataset for pixel-based image classification in google earth engine *Remote Sens.* **8** 634

Gumbricht T, Roman-Cuesta RM, Verchot L, et al. An expert system model for mapping tropical wetlands and peatlands reveals South America as the largest contributor. *Glob Change Biol*. 2017; 23: 3581–3599. https://doi.org/10.1111/gcb.13689

Garcia M 2021 *Investigating the Use of Spatially-Explicit Modelling and Cross-Validation Strategies in Spatial Interpolation Machine Learning Problems* available at: https://run.unl.pt/handle/10362/113881

Goldblatt R, You W, Hanson G and Khandelwal A K 2016 *Detecting the boundaries of urban areas in india: a dataset for pixel-based image classification in google earth engine Remote Sens.* **8** 634

Hastie et al 2024. *A new data-driven map predicts substantial undocumented peatland areas in Amazonia*. Environ. Res. Lett. 19 094019. https://iopscience.iop.org/article/10.1088/1748-9326/ad677b

Meyer H and Pebesma E 2022 *Machine learning-based global maps of ecological variables and the challenge of assessing them Nat. Commun.* **13** 2208